# A Guide to Your Desired Lipid-Asymmetric Vesicles

**DOI:** 10.3390/membranes13030267

**Published:** 2023-02-23

**Authors:** Mona Krompers, Heiko Heerklotz

**Affiliations:** 1Department of Pharmaceutical Technology and Biopharmacy, Institute for Pharmaceutical Sciences, University of Freiburg, 79104 Freiburg im Breisgau, Germany; 2Leslie Dan Faculty of Pharmacy, University of Toronto, Toronto, ON M5S 3M2, Canada; 3Signalling Research Centers BIOSS and CIBSS, University of Freiburg, 79085 Freiburg im Breisgau, Germany

**Keywords:** lipid asymmetry, lipid exchange, liposome preparation, model membrane, phospholipids, cholesterol, cyclodextrin, emulsion phase transfer, microfluidics

## Abstract

Liposomes are prevalent model systems for studies on biological membranes. Recently, increasing attention has been paid to models also representing the lipid asymmetry of biological membranes. Here, we review in-vitro methods that have been established to prepare free-floating vesicles containing different compositions of the classic two-chain glycero- or sphingolipids in their outer and inner leaflet. In total, 72 reports are listed and assigned to four general strategies that are (A) enzymatic conversion of outer leaflet lipids, (B) re-sorting of lipids between leaflets, (C) assembly from different monolayers and (D) exchange of outer leaflet lipids. To guide the reader through this broad field of available techniques, we attempt to draw a road map that leads to the lipid-asymmetric vesicles that suit a given purpose. Of each method, we discuss advantages and limitations. In addition, various verification strategies of asymmetry as well as the role of cholesterol are briefly discussed. The ability to specifically induce lipid asymmetry in model membranes offers insights into the biological functions of asymmetry and may also benefit the technical applications of liposomes.

## 1. Aims and Content of This Review

For decades, it has been known that most biological, lipid-bilayer based membranes are asymmetric in containing other lipids in the outer than in the inner membrane leaflet [1]. The considerable effort of an organism to establish, maintain, and adapt this asymmetry implies important biological functions [2]. However, since virtually all model membranes used in biophysical and biochemical studies were symmetric, these functions have remained largely unclear. Over the last few years, this long-term shortcoming has been overcome by a large-scale effort to establish and apply new, asymmetric membrane models. 

Our review of this highly dynamic field has two main aims. First, we attempt at compiling all assays and protocols to prepare free-floating, lipid-asymmetric vesicles of the classic two-chain, glycero- or sphingolipids reported so far. Table 1 compiles the impressive number of 72 reports differing in strategy or lipid composition that we were able to find. Second, our paper aims at sorting these strategies and protocols into different principal categories and offering a road map that might help with finding the right protocol for a given purpose. For the sake of keeping this paper short and focused, we excluded other, certainly also very interesting membrane models such as asymmetric black lipid membranes, droplet interface bilayers, supported lipid bilayers, multicompartment vesicles, hybrid polymer-lipid vesicles and plasma membrane vesicles, and we did not list work on other lipidic compounds such as ceramides, gangliosides, lyso-lipids or lipopolysaccharides. Finally, we compile the applications as reported, for example a method demonstrated for asymmetric giant unilamellar vesicles (aGUVs) only, and abstain from speculation whether and how existing methods could be adapted or developed to serve other purposes in the future. Of course, such developments are expectable.

Excellent, alternative reviews that focus on other aspects of the field are available. Some articles address the production and application of GUVs in particular [3,4,5,6,7]. Dimova et al. focused on the preparation of aGUVs, in particular their observation by optical microscopy [3]. Reports about the preparation of aGUVs also include various microfluidic-based technologies [5,6,7,8,9]. Huang et al. described microfluidic emulsification in terms of microfluidic fabrication of single, double, triple or higher-order emulsion drops [8]. Kamiya and coworkers [5,6] discussed several techniques based on microfluidics for GUV formation. They summarized the properties of each method, including effects on encapsulation efficiency, size range and asymmetry of membranes. In addition, they described the formation of complex structures in terms of fabricating artificial cell models [5,6]. Cespedes et al. [10] reviewed the interplay between membrane components and the physical properties of the plasma membrane. Their report is outstanding for its focus on the immunological synapse [10]. London and coworkers mainly reviewed cyclodextrin-based methods for preparing asymmetric liposomes [11,12]. Besides this, Kakuda et al. [11] summarized studies about pore-forming toxins, such as perfringolysin O (PFO), regarding lipid interactions in symmetric and asymmetric vesicles [11]. In another article, studies about the effects of asymmetry on the ability of membranes to form ordered domains are summarized [12]. Scott et al. [13] recently reviewed experimental and computational techniques to study membrane asymmetry. Their focus was on in vitro methods that have advanced the understanding of the plasma membrane, along with molecular dynamics simulations. Different techniques for the fabrication of large and giant vesicles are described, i.e. via Ca^2+^-ions, enzymes and cyclodextrins. With respect to GUV preparation, i.e. hemifusion and phase-transfer approaches are described [13].

Overviewing the different strategies compiled in Table 1, we state that most start with symmetric vesicles and render them asymmetric in another preparation step. This may be achieved by enzymatic conversion of one lipid species into another one (A), by inducing the flip or flop of a given lipid species to accumulate in one leaflet (B), or by exchanging lipids in the outer leaflet (D). A fundamentally different approach is to assemble the vesicle bilayer from individual monolayers from scratch (C). These four fundamental strategies are pursued by many different protocols which all have their specific requirements, limitations, benefits and drawbacks.

## 2. Navigating the Preparation of Asymmetric Model Membranes

We provide a map to help navigate through this field to find a suitable preparation method that meets individual requirements and possibilities especially in terms of practical implementation (see Figure 1). If you have decided to use asymmetric vesicles but have not made up your mind regarding the specific protocol, “you are here” on the left side of the map and get going down Main Street. On your way, there will be exits to different methods that may or may not be available and favorable for you. As an alternative, you can always stay on Main Street.

For your choice, you will need to rank your options to optimally suit your purpose with respect to the degree of asymmetry, reliability, stability, absence of disturbing components and the time, equipment, materials and expertise needed. Our review cannot solve this problem for you. We are citing some available information on these points given in the original papers but we are lacking a true comparison of different protocols done by the same lab. Furthermore, the best choice depends on the problem and the equipment and experience of a given laboratory. In other words, we try to provide a map but your best path will depend on whether you are driving a racecar, a 4 × 4 or a bicycle.

### 2.1. Enzymatic Conversion of Outer Leaflet Lipids

The first option to turn off Main Street is taking a right on Enzyme Road. You may take this exit if you are lucky to have an enzyme available that locally (typically in the outer leaflet) converts an undesired lipid into a desired lipid. This is very advantageous for example if you aim at a limited amount of a lipid in the inner leaflet only. Eliminating a minor component in the outer leaflet by unspecific exchange would require the complete replacement of all outer leaflet lipids. The specific elimination of the minor fraction only is much more elegant and less harsh to the vesicles [14]. If such an enzyme is not available or favorable, you may skip this section and stay on Main Street.

To our knowledge, two enzymatic methods have been developed so far, using a decarboxylase (A1) or phospholipase (A2). The approach allows for a minimal invasive formation of asymmetric lipid distributions in the vesicle bilayer, leaving other lipids unaffected [15].

Phosphatidylserine decarboxylase (PSD) converts only phosphatidylserine (PS) localized in the outer leaflet into phosphatidylethanolamine (PE) in only a few steps [16]. We previously described a protocol to produce “20 mol% PS inside” liposomes in only one or two hours, with high but not complete asymmetry and controlled composition that mimics the PS asymmetry of mammalian cell membranes [15].

Phospholipase D (PLD) hydrolyzes phospholipids to phosphatidic acid (PA), whereby the converted lipids can be head group-labeled fluorescent phospholipid analogues [17]. PLD also promotes the transphosphatidylation of phosphatidylcholine (PC) to PS and PE in the presence of serine and ethanolamine [14]. Asymmetric liposomes with about 95% of PC molecules localized to the inner leaflet [14], i.e., 49% conversion of total phospholipids [17] can be produced. The substrate specificity of enzymes, however, limits these methods to specific types of lipids [14]; therefore, it restricts the variety of lipid species that can be asymmetrically distributed in the bilayer. Complete enzymatic lipid conversion presents a difficulty [15], which is why full asymmetry cannot be achieved [14]–[17].

### 2.2. Re-Sorting of Lipids between Leaflets

The second, rather small lane branching off Main Street to the right is Flip Lane, leading to methods B1 and B2. In the case that a desired lipid allows for forced flip, re-sorting of lipids between leaflets via pH gradients (B1) or via Ca^2+^-ions (B2) can produce lipid-asymmetric liposomes.

The accumulation of weak acids or bases on one side of a membrane by gradients of pH or complexing agents has long been used to achieve extreme encapsulation efficiencies for the liposomal delivery of water-soluble drugs [18]. A limitation for membrane lipids is that, in contrast to exchanging one lipid for another, a directed transfer of lipids creates an imbalance between the intrinsic areas of the leaflets, i.e., asymmetry stress. If this issue cannot be dealt with specifically, this method must be limited to a very small fraction of a lipid in the membrane.

Weakly acidic lipids can be sorted using the pH gradient method: since lipids may flip across a membrane only in the neutral, but not in charged form, they will accumulate on the low-pH side where they get charged and membrane-impermeant [19]. For anionic lipids, net transport then proceeds from the low-pH side of the bilayer to the high-pH side [20]. This method involves only a few preparation steps such as buffer change, initiation and stop of lipid transport [19,20,21,22,23,24,25]. The pH gradient-induced generation of asymmetric vesicles has been used to modulate membrane fusion [20] and to prepare aGUVs [24]. Certain factors can influence the generation of asymmetry, such as negative surface charge on the membrane, lipid saturation or addition of cholesterol [23]. The amount of lipid transported is limited: for instance, transport of 5% of the total outer leaflet lipid to the inner leaflet has been reported. However, lipid redistribution can occur extremely rapidly [22]. Caution is advised when interpreting the degree of asymmetry specified in such systems. Imagine a vesicle containing 1 mol% of lipid X that is treated in a way to accumulate 95% of this lipid X in the inner leaflet. The resulting vesicle can be referred to as “95% asymmetric with respect to X” but is about 2% asymmetric overall (0.1 mol% of X in outer and 1.9 mol% in inner leaflet) [19,21,22,23,25].

Sun et al. [26] developed a protocol to re-sort PS lipids via ions: the presence of Ca^2+^-ions combined with an incubation temperature of 70 °C for a certain time allows for controlled production of PS-asymmetric vesicles [26]. The low Ca^2+^ concentration in the core of the vesicle lets the complex dissociate and entraps the PS. Guo et al. [27] showed that PS flip to the inner membrane leaflet is affected by vesicle size, incubation temperature and lipid composition. Particularly vesicle size and PS content affect the formation of asymmetric lipid distribution, which permits regulating the degree of asymmetry of PS-containing vesicles. Asymmetry remains for days due to a lower activation energy of the flip process compared to the flop process when incubated with Ca^2+^. However, using vesicles of 400 nm size slows down the formation process of asymmetric vesicles compared to smaller sizes (50 nm). With increasing PS contents of the vesicles, maximal asymmetry decreases [27]. Note that the thermal stability of lipids should be considered. Further, both methods are limited with respect to lipid variety: pH gradients can only induce asymmetric distribution of phospholipids which are weak acids [21], whereas the Ca^2+^ method is specifically applicable to PS lipids [26].

### 2.3. Assembly from Monolayers

The first two exits to the left belong to the ‘assembly from monolayers’ county. It is, of course, fascinating to put together a custom-designed asymmetric vesicle directly. One price to pay for this is the involvement of an organic phase that typically leads to more or less organic solvent to remain in the final vesicles (see below). Micro Road is a fairly fast and fancy way to microfluidic technologies which prepare lipid-asymmetric vesicles either via inkjet printing (C3) or via the droplet transfer method (C2). As of today, these instrumentations are not lab standard, so it is a toll road.

Giant Street to droplet transfer will lead you to aGUVs only (C1), but is easier and cheaper to travel. The droplet transfer method originally established by Pautot et al. [28,29] involves, first, the introduction of water droplets into an organic solution of the lipid desired to form the inner leaflet. Spontaneously, a monolayer of lipid covers the water droplets with the chains reaching to the outside (water-in-oil: w/o emulsion). Then, the droplet is forced to cross a boundary from the organic solvent to water, which is covered with the lipid needed for the outer leaflet. The lipid of the surface film will surround the droplet to make its outer surface hydrophilic. Moving micron-sized droplets across the boundary by centrifugation produces aGUVs in the aqueous phase [28]. Using microfluidic technologies, the inner leaflet of the membrane can be prepared by injecting finely tuned water droplets one by one into a flow of a continuous oil phase and then leading them to become surrounded by the water phase [30]. Modifications in microfluidics include layer-by-layer membrane assembly [31], double emulsion [32] and triple emulsion techniques [33], as well as polycarbonate filter systems [34] and dielectrophoretic separation of microemulsions [35]. The inkjet printing method starts from a planar, asymmetric bilayer formed at the contact of two aqueous droplets in an oily phase. From this bilayer, vesicles are ejected by the printing pulse [36,37,38,39,40].

Arriaga et al. [33] summarized various aspects of some of the protocols shown here, including time stability of asymmetry as well as advantages and disadvantages. The production of vesicles via inverse emulsion or droplet phase transfer without using microfluidic devices is easy to implement and leads to high asymmetries up to 95% [33]. Hamada et al. [41] provided a centrifuge-independent method with real-time observation of the transfer process. Vesicle size can be adjusted via sugar gradient [41]. However, the method is limited to low encapsulation and throughput. Vesicle size in general is difficult to control, leading to polydisperse sizes [33]. The phase transfer method is incompatible to lipids that display poor solubility in oil due to their net charge or saturated fatty acid tails [36].

Advantages of microfluidic technologies include high encapsulation efficiency, control over lamellarity and monodisperse vesicle sizes [30,31,32,33,34,35,42,43]. Single-chip microfluidic platforms combining several fabrication steps allow for high-throughput liposome production [32,33,43]. Inkjet printing is also applicable to lipids with poor solubility in oil [36] and achieves long-term stability of at least seven days [37]. However, it requires more specialized equipment than other approaches [36]. Yet, a limited number of solvents can be applied when using poly(dimethylsiloxane) (PDMS) based microfluidic devices [31,32,43].

In general, assembly from different monolayers and control over the composition of each leaflet with or without microfluidic devices leads to high asymmetries up to 100% [28,30,31,32,33,34,35,42]. It enables the encapsulation of macromolecules at any concentration and the use of a wide variety of lipids [28,30,36]. As the lipids are dissolved in organic solvents, oil residues are trapped in the bilayer, possibly affecting membrane properties [28,33,41]. Therefore, some protocols aim to minimize such oil residues [35,36]. If it cannot be avoided to use organic solvents within the vesicle formation procedure, it should be tested if such oil contaminants affect lipid or membrane properties. For instance, Elani et al. [44] studied mechanical properties in terms of vesicle bending rigidities and concluded that the entrapped oil does not influence the above mentioned properties of the membrane [44].

### 2.4. Exchange of Outer Leaflet Lipids

If none of the exits offered so far turned out to be accessible and particularly attractive, what is left is a set of techniques having in common the exploitation of lipid exchange. As of today, exchange of outer leaflet lipids seems to be the most versatile and widely used strategy able to tackle virtually every lipid asymmetry. Of course, this group of methods does not come without drawbacks and limitations, too. Let us, first, give an overview of the options.

Users aiming at aGUVs who did not exit to the emulsion-based methods (C1, C2) before may go for hemifusion-based exchange (D1) with an excess area of solid supported membranes (see also [45,46,47]). This approach works without a lipid carrier and the elimination of the donor reservoir after exchange is straightforward.

Introducing a defined amount of a component to the outer leaflet of large unilamellar vesicles (LUVs)—such as 20 mol% phosphatidylglycerol (PG) mimicking this asymmetry of bacterial membranes [48]—without the need for donor aggregates has been done by exchange between the liposomes and cyclodextrin-solubilized lipid in aqueous solution (D2). It has also served for forming lipid-asymmetric proteoliposomes [49,50]. An advantage compared to the exchange with donor liposomes or bilayers (D1, D3, D4, D5) is that all of the donor lipid equilibrates very quickly with the outer leaflet of the acceptor vesicles. Application requires knowledge of the cyclodextrin concentration needed to fully solubilize a certain amount of donor lipid.

The lane to protein-mediated lipid transfer (D3) has been little travelled and maintained recently, maybe since cyclodextrins are much simpler and more versatile, but may offer interesting future applications.

Solid-supported vesicles had been used for the Transil^TM^ partitioning assay [51] and to render solid-supported lipid bilayers asymmetric [52]. More recently, they were used as donors for producing lipid-asymmetric liposomes (D4) [53]. In addition to offering an elegant solution to eliminate donors, they also activated exchange without a carrier by increasing the temperature. Naturally, this route should only be travelled for thermally stable lipids and liposomes with asymmetry of sufficient thermal stability.

It reflects our personal view of the current literature of the field that whoever did not have a chance or did not bother to exit Main Street to explore potentially advantageous sideroads will finally cross London Bridge to the protocol of lipid exchange between donor and acceptor vesicles.

Let us address the methods D1-D5 in some more detail. Enoki et al. [45] established the preparation of aGUVs via hemifusion of symmetric GUVs with a solid supported bilayer (SLB). In the hemifusion state induced by the presence of Ca^2+^, GUVs dock to the support and outer leaflet lipids exchange by diffusion. By eliminating Ca^2+^ using a chelator, hemifusion is reversed and the aGUVs detach from the SLB [45]. By using the hemifusion method, the resulting aGUVs are free of any exogenous contaminants such as cyclodextrins or organic solvents, except for trace fractions of fluorescence dyes used for detecting asymmetric exchange. Preparation of aGUVs and data collection needs less than five hours. The resulting vesicles show high asymmetries approaching 100%, given the large excess of the SLB donor area over the GUV area [45,46,47]. However, during the hemifusion process or when aGUVs are sheared off the SLB, transient pores are formed so that lipid flip-flop may occur. Such “leaky” GUVs should be identified and excluded from subsequent aGUV experiments [45].

Using solubilized donor lipids (D2) [48] rather than donor vesicles or bilayers has two main advantages. Exchange is very fast and complete, yielding a well-defined content of donor lipid in the target liposomes. Second, the elimination of the cyclodextrin complexes after exchange is either unnecessary, if subsequent experiments are not compromised by ongoing equilibrium exchange, or trivial. This protocol was also utilized for the preparation of lipid-asymmetric proteoliposomes, containing a large, multi-spanning membrane protein, the antiporter ST-NhaA [49] and the ligand-gated ion channel ELIC [50]. By now, this has been demonstrated only with very few procedures [49,50,54,55].The principle of this approach is to first completely dissolve donor liposomes to obtain a solution of mβCD−donor lipid (in our example, PG) complexes. This solution is then equilibrated with a proper amount of “acceptor” liposomes so that PG enters the membrane and the corresponding amount of PC is solubilized instead. To determine the required amounts of lipid and cyclodextrin for the desired degree of lipid exchange, lipid-cyclodextrin interactions were previously investigated by isothermal titration calorimetry (ITC) experiments; an alternative method would be light scattering (see below). Within one “round” of exchange, 5 to 45 mol% lipid were exchanged in the outer vesicle leaflet. The asymmetry of LUVs remained stable for 14 days [48] and of proteoliposomes for seven days [49]. A detailed protocol and Excel^TM^ sheet is provided to calculate required lipid and mβCD concentrations [48].

Living organisms transport lipids by a variety of specific carrier proteins that may be used to facilitate exchange without the need for cyclodextrins. PC molecules, for instance, were introduced into the outer vesicle monolayer via exchange protein from bovine liver [56]. PC-specific exchange protein was also used to prepare vesicles with an asymmetric distribution of brominated PC molecules. However, brominated PC lipids possibly have adverse effects upon the enzymatic activity of some reconstituted systems in model membranes [57]. Sandra et al. [58] generated PE-asymmetric vesicles by incubating lipid vesicles with rat liver exchange protein and a suitable acceptor membrane. Only the outer surface of PE-containing vesicles is accessible to the exchange protein, which leads to an asymmetric lipid distribution across the bilayer [58]. Holzer et al. [59] initiated protein-mediated lipid transfer between egg-PC (EPC) acceptor vesicles and EPC:EPG 90:10 mol% donor vesicles with the help of recombinant pro-sterol carrier protein 2 (pro-SCP2). Using this protein-mediated strategy for lipid exchange, aLUVs were fabricated in less than three hours. As a result, the amount of EPG in acceptor vesicles increased to 3 mol%, whereas EPG in donor vesicles was reduced to 6 mol%. Pro-SCP2 accelerates the EPG transfer to half-times of between two and three hours, and thus, minimizes lipid flip-flop during the transfer process. In comparison, the spontaneous redistribution of EPG occurs at half-times of tens of hours. Note that liposome size affects the degree of asymmetry. Narrow size distributions are important for obtaining aLUVs with a uniform degree of asymmetry [59]. In general, the application of protein-mediated lipid exchange is limited to a few lipid species. Further, post-exchange acceptor vesicles can be contaminated with donor vesicles [56]. Unless the use of cyclodextrin has to be avoided in a given system or for a given experimental technique, there seems to be little motivation to use a transfer protein in a rather unspecific manner. The true potential of this approach would be to selectively add or remove a component while leaving all others unaffected, as discussed above for enzymatic conversion. However, specific insertion or extraction of a single lipid without creating or filling the “gaps” with another would create asymmetry stress as discussed in the section on induced flip. Reaching substantial asymmetries by selective transfer would require a solution to this problem.

Another possibility to exchange lipids of the outer leaflet is using solid-supported nanoparticles as the donor phase, and a high temperature to activate exchange [53]. Small unilamellar vesicles (SUVs) of desired composition were prepared by adjusting parameters such as temperature, time and ratio of lipid-coated silica nanoparticles to vesicles. The use of lipid-coated nanoparticles facilitates the purification process for the easy preparation and isolation of asymmetric vesicles. Here, lipid exchange proceeds at 75 °C [53]. Note that elevated temperatures activate both the desired lipid exchange and the detrimental intra-bilayer lipid flip-flop in asymmetric liposomes [60]. Hence, the exchange protocol has to find a compromise between these effects: the authors managed to reach a final content of donor lipid, 1,2-dipalmitoyl-sn-glycero-3-phosphocholine (hDPPC), of 20 mol% in the outer compared to 5 mol% in the inner leaflet. The method was demonstrated for saturated, thermally stable, isotopically distinct DPPC lipid molecules [53].

The exchange between donor and acceptor liposomes is a classic and most widely used method established by London and coworkers. The principle of this method is to facilitate the exchange between donor and acceptor vesicles by relatively small concentrations of cyclodextrin which solubilize only a little lipid at a time but shuttle some of it between the vesicles. After exchange, the now-asymmetric acceptor vesicles need to be separated from a potentially large excess of donor vesicles by centrifugation [61,62,63]. Over the last decade, several modifications of this method have been developed, which differ in, for instance, cyclodextrin species, vesicle size and centrifugation procedures. Besides methyl-β-cyclodextrin (mβCD) [55,60,61,62,63,64,65,66,67,68,69,70,71,72,73,74,75,76,77], hydroxypropyl-α-cyclodextrin (HPαCD) [78,79,80,81,82] and methyl-α-cyclodextrin (mαCD) [83,84] was also used. Mainly aLUVs were produced, but some protocols resulted in fabricating aSUVs [61,63,75,76] and aGUVs [55,67,73,75,80]. Two main procedures have been established regarding the centrifugation step for vesicle separation: the heavy-acceptor (ha) and heavy-donor (hd) strategy. Cheng et al. first provided the ha-strategy [61], which was modified later by Heberle et al. [62] in terms of loading donor vesicles instead of acceptor vesicles with sucrose solution. Doktorova et al. [63] provided a detailed protocol, including the two strategies mentioned above. In about 12 h, this protocol can produce up to 20 mg of asymmetric vesicles. Comparing hd- and ha-strategies, the latter simplifies purification, whereas the hd-strategy excludes sucrose from acceptor vesicles. For the hd-strategy, additional purification steps may be required depending on the density of the donor lipid, resulting in reduced yield. Moreover, entrapped sucrose induces osmotic stress, potentially causing bilayer thinning and lipid area expansion [63]. An approach of Li et al. [84] entraps physiological osmolalities of cesium chloride (CsCl) inside acceptor aLUVs instead of sucrose. The density of liposomes is increased without the use of a hypertonic sucrose solution, preventing acceptor vesicles from osmotic pressure imbalance. CsCl entrapment did not interfere with the ability to produce aLUVs or maintain efficient exchange [84].

**Table 1 membranes-13-00267-t001:** Summary of protocols for in-vitro preparation of lipid-asymmetric vesicles. Articles are listed by category (A–D) and date of publication. Selected features are shown including vesicle type, degree of asymmetry (asy), outer leaflet and inner leaflet composition, asymmetry verification method and a short description of the respective article. Lipids primarily intended to be asymmetrically present in one leaflet are highlighted in bold. Lipids in light font are matrix or acceptor lipids that may be present in both leaflets. Note that the degree of asymmetry is interpreted in different ways; thus, footnotes are inserted to provide more detailed information.

Vesicle Type	Asy	Outer Leaflet	Inner Leaflet	Verification of Asy	Short Description	Ref.
A. Enzymatic conversion of outer leaflet lipids						
A.1. Decarboxylase						
LUV	97% ^1^	DOPC/NBD-**PE**	NBD-**PS**	FRET, trinitrophenylation	one-step method, enzyme conversion of PS to PE by PS-decarboxylase	[16]
LUV	a = −0.5 (PS), a ≈ 1 (PE) ^2^	ePC/**PE** PC/chol/eSM/**PE** PC/**PE** PC/PE/PG/**PE**	**POPS** **POPG** **POPS/POPG**	ζ-potential, HPTLC	PS-decarboxylase converts PS to PE, aLUVs mimic PS-asymmetry of eukaryotic plasma membranes	[15]
A.2. Phospholipase D						
LUV	49% ^1^	**PA**	PC/PE/N-NBD-PE/N-Rho-PE	F (N-Rho-PE, N-NBD-PE)	outer lipid conversion to PA, influenza-induced fusion between viral and liposome membrane	[17]
LUV	>95% ^3^	**POPS/POPE**	POPC± chol	enzymatic assay/optical absorption, HPLC	enzymatic conversion of PC in the presence of serine and ethanolamine	[14]
B. Re-sorting of lipids between leaflets						
B.1. pH gradient						
LUV	80–90% ^10^	DOPC	**ePG** **DOPA**	ion-exchange C, ^13^C NMR, periodate oxidation	asymmetric distributions of PA in aLUVs via pH gradients	[21]
LUV	50% ^13^	DPPC DPoPC DOPC ePC ePC/chol ePC/PS	**ePG** **DOPG** **MOPG**	periodate oxidation	mechanism of pH-induced PG trans-bilayer transport	[22]
LUV	>80%	**PA** PC **CL**	PC **PG** **SA**	two-phase polymer partition, ^3^H-radioactivity	effect of temperature and lipid composition on formation and extent of asymmetry	[23]
LUV	>95% ^13^	**DOPA** DOPE/DOPC/PI	DOPE/DOPC/PI **DOPA**	F (TNS)	influence of lipid asymmetry on Ca^2+^-stimulated vesicles fusion	[19]
GUV	n.a.	**ePG**	ePC	phase contrast M	influence of lipid redistribution on the shape of GUVs	[24]
LUV	n.a.	ePC/chol ePC/DOPE/chol	**amino lipids AL1-AL6**	F (TNS)	pH gradient induced fusion of liposomes containing synthetic amino lipids	[20]
LUV	>80% ^13^	DOPC ePC	**DOPA** **ePA**	NMR	NMR observation on transbilayer distribution of Chlorpromazine	[25]
B.2. Ca^2+^-ions						
LUV	≤30% ^13^	DPPC	**DOPS**	FQ (NBD-PS), nanoDSC	Ca^2+^-induced inward flip of PS for controlled production of aLUVs	[26]
LUV	38.5–52.3% ^14^	DPPC	**DOPS**	FQ (NBD-PS), nanoDSC	effect of size, temperature and lipid composition on Ca^2+^-induced PS inward flip	[27]
C. Assembly from monolayers						
C.1. Droplet transfer/emulsion phase transfer						
GUV	≤95%	**POPC** **ePC**	**POPS** **polystyrene-polyacrylic acid**	FQ (NBD-PE, NBD-PS)	engineering aGUVs with two independently prepared monolayers	[28]
GUV	n.a.	**POPC/py-16-PC** **POPC/POPE/CL**	**POPC/POPE/CL** **POPC/py-16-P**	F (pyrene)	membrane − protein interactions between Bax and liposomes of size 0.3-1.5 µm	[85]
GUV	n.a.	**DOPC**	**ePC** **DOPC/DPPC/chol**	FM (Rho-PE, NBD-PE)	cell-sized aGUVs, control over vesicle size via sugar gradient	[41]
GUV	n.a.	**ePC** **DOPG**	**DOPE** **ePC** **DOPG**	n.a.	reconstitution of the potassium channel KcsA into aGUVs	[54]
GUV	n.a.	**POPC** **DOPC**	**DOPC** **POPC**	FM of hemifused GUVs (Rho-PE, NBD-PE)	effects of lipid asymmetry on membrane bending rigidity	[44]
GUV	n.a.	**DOPC/chol** **NBPC/chol**	**NBPC/chol** **DOPC/chol**	FM (Rho-DHPE)	asymmetric distribution of photocleavable lipid, photoinduced pinocytosis behaviour	[86]
GUV	n.a.	**DSPE/DSPG**	**DSPG** **DSPE** **DOPG** **DOPC**	FQ (NBD-PE)	influence of lipid head group and acyl chain on Daptomycin-induced membrane permeability	[87]
GUV	n.a.	**DOPC** **DOPC/DOPS** **DOPC/DOPG**	**DOPC/DOPS** **DOPC/DOPG** **DOPC**	F Annexin V (Alexa Fluor 488)	protein translocation via cell-penetrating peptides, start of enzymatic reactions in aGUVs	[88]
C.2. Droplet transfer/microfluidic technologies						
GUV	85%	**DPPC** **DOPC** **PS**	**DPPC** **DOPC** **PS**	FQ (Texas Red (TR)-modified DPPE), biotin-binding (biotin-DPPE, avidin), F Annexin V (Alexa Fluor 488)	two-step fabrication of monodisperse and unilamellar aGUVs	[30]
GUV	100%	**NBD-DOPC**	**DOPC**	F (NBD-DOPC)	controlled construction of uni- or multilamellar aGUVs using layer-by-layer membrane assembly	[31]
GUV	90–95%	**DOPC**	**DOPE**	FM, FQ (NBD-DOPC, TR-DOPE)	continuous fabrication of aGUVs via double emulsions with customized membrane composition, size and luminal content	[32]
GUV	95%	**DMPC** **DOPC**	**DOPC** **DMPC**	FQ (NBD-PC)	influence of asymmetry on area expansion modulus, customized micropipette aspiration system	[42]
GUV	n.a.	**DOPC**	**POPC**	F click chemistry (DSPE-DBCO, 3-azido-7-hydroxycoumarin)	high-throughput fabrication of aGUVs from aqueous lipid dispersions	[43]
GUV	≤70%	**DOPC** **DOPE-biotinyl**	**DOPC** **DOPE-biotinyl/DOPC**	F/biotin-streptavidin (DOPE-biotinyl, streptavidin fluorescein isothiocyanate ST-FITC)	continuous single-step fabrication in a glass device using triple emulsion drops	[33]
GUV, LUV	79%	**POPS**	**POPC**	FM (carboxy-fluorescein), FQ (NBD-PC)	preparation of liposomes with desired diameters using a tunable microfluidic device including a polycarbonate filter	[34]
GUV	83%	**POPC**	**DOPC**	FQ (NBD-DHPE)	aGUVs with precisely modulated size and minimal oil contamination	[35]
C.3. Inkjet printing						
GUV	n.a.	**DPhPC/TMR-PIP2** **PE-PEG2000** **DPhPC/DGS-NTA-Ni** **DPhPC** **DPhPC/DPhPS/chol**	**DPhPC/TMR-PIP2** **PE-PEG2000** **DPhPC/DGS-NTA-Ni** **DPhPC** **DPhPC/DPhPS/chol**	FM (TMR-PIP2, His-GFP)	separate vesicle and bilayer formation allows for monitoring and minimizing oil contamination	[36]
GUV	n.a.	**DOPC** **DOPS**	**DOPC** **DOPS** **DOPS/DOPE/DOPC**	F Annexin V (Alexa Fluor 488, 546)	membrane dynamics and protein interactions, use of little organic solvent	[37]
GUV	n.a.	**DOPC** **DOPC/DOPE**	**DOPC/DOPS**	FQ (Rho-DOPE)	device for sequentially generating aGUVs, influence of the peptide Cinnamycin on lipid dynamics	[38]
LUV	n.a.	**DOPC** **biotin-DOPE**	**biotin-DOPE** **DOPC**	Biotin-streptavidin (biotin-PEG(2000)-DSPE, Streptavidin-conjugated gold) and TEM, FM (Rho-DOPE)	fabricating nano-sized liposomes from a planar lipid bilayer by applying a pulsed-jet flow with optimized duration and pressure	[39]
GUV	n.a.	**DOPC** **DOPC/DOPS**	**DOPC/DOPS** **DOPC/DOTAP**	FM (Rho-DOPE)	fusion between LUVs and a monolayer, followed by application of a pulsed jet flow	[40]
D. Exchange of outer leaflet lipids						
D.1. Hemifusion						
GUV	50–99% ^12^	DOPC/chol	**DSPC**	FM (TRPC, DiD), FQ (NBD-PE)	hemifusion of giant vesicles and a supported lipid bilayer	[45]
GUV	≤86% ^12^	DOPC/chol	**DSPC/POPC**	FM (TRPC, DiD)	systematic study of aGUVs to investigate modulated phases	[46]
GUV	>70% ^12^	DOPC DOPC/chol	**DSPC** **bSM** **DSPC**	FM (TRPC, DiD)	phase behavior and cholesterol movement in aGUVs	[47]
D.2. Complexes in aqueous solution						
LUV	0.05–0.45 ^9^	**POPG**	POPC	ζ-potential	PG-loaded mβCD-lipid-complexes in solution replace PC by PG	[48]
LUV	0.2 ^9^	**POPG**	POPE/POPC/TOCL	ζ-potential	five-step protocol to proteoliposomes with incorporated ST-NhaA	[49]
LUV	0.25 ^9^	**POPG**	POPC POPC/POPE	ζ-potential	phospholipid modulation of ELIC in PG-asymmetric proteoliposomes	[50]
D.3. Protein-mediated lipid transfer						
SUV	10–20% ^4^	**[N-^13^CH_3_]-DOPC** **[N-^13^CH_3_]-DMPC**	DMPC DOPC	^13^C-NMR	transfer of PC between acceptor and donor PC vesicles	[56]
MV	62% ^4^	**[^3^H]-DOPC** **[^3^H]-DOPE**	DOPC/DOPE/CL	radioactivity, TNBS-labeling	lipid transfer between isotopically asymmetric vesicles and chinese hamster fibroblasts	[58]
SUV	≤59.1% ^4^	**BRPC** **POPC**	POPC BRPC	F (CUGA), GC	studies on the membrane-binding domain of cytochrome b5 in brominated aSUVs	[57]
LUV	60% ^13^ or 3 mol% ^5^	**ePG**	ePC	FFE	pro-SCP2 mediated EPG transfer, separation of donor and acceptor vesicles via FFE	[59]
D.4. Solid-supported nanoparticles						
SUV	75.6 mol% ^5^, 24.4 mol% ^10^	**hDPPC**	d62DPPC d75DPPC	SANS, ^1^H-NMR	lipid exchange via lipid-coated silica nanoparticles	[53]
D.5. Donor liposomes and cyclodextrin						
SUV	75–82% ^4^	**bSM**	DOPC POPC POPS/POPE DOPC/chol POPE/POPS/chol	FA (DPH, TMA-DPH), HPTLC, pL4A18 peptide binding	mβCD-mediated lipid exchange, ha-strategy	[61]
GUV	60% ^4^	**bSM**	DOPC bPC bPC/bPE ± chol	FCS (Nile-red NR12S, NBD-PE)	solvent free method for mβCD-induced lipid exchange to prepare aGUVs, ha-strategy	[73]
LUV	80–100% ^4^	**bSM** **bSM/POPC**	DOPE/POPS ± chol	FA (DPH, TMA-DPH), HPTLC, pL4A18 peptide binding	mβCD-mediated exchange (ha-strategy), investigation of interleaflet coupling	[74]
GUV, SUV	20–80 mol% ^4^	**bSM** **mSM** **C24:0-SM/bSM**	DOPC POPC/bSM SOPC/bSM OMPC/bSM	FLIM (NBD-DPPE, NBD-DOPE, TMA-DPH)	mβCD-induced exchange (ha-strategy) with lipids of various acyl chains, investigation of interleaflet coupling	[75]
SUV	62–96% ^4^	**bSM**	di14:1PC di16:1PC di18:1PC di20:1PC di22:1PC diphyPC 16:01-18:2PC 16:0-20:4PC di18:2PC di18:3PC di20:4PC	FA (TMA-DPH), HPTLC	mβCD-mediated lipid exchange (ha-strategy), effects of PC acyl chain structure	[76]
LUV	>90% ^4^	**bSM** **PC** **bSM/PC**	PE/PS/chol	HPTLC, pL4A18 peptide binding, TNBS-labeling	HPαCD-mediated exchange (ha-strategy) with controlled amount of cholesterol (0-50 mol%)	[78]
LUV	80–90% ^4^	**POPC**	POPE/POPS/chol	HPTLC, pL4A18 peptide binding, TNBS-labeling	HPαCD-mediated exchange (ha-strategy), influence of lipid composition and asymmetry on Perfringolysin O	[79]
GUV, LUV	50 mol% ^5^	**eSM** **mSM**	DOPC/chol	F (Rho-DOPE, NBD-DOPE), FA (TMA-DPH), HPTLC	solvent-free method for HPαCD-induced lipid exchange (ha-strategy) and control of cholesterol	[80]
LUV	n.a.	**bSM**	DOPE/POPS	F (Topfluor-PC)	mβCD-induced exchange (ha-strategy), antibody-decorated aLUVs bind HIV-1 virus-like particles	[77]
LUV	≤0.95 ^6^	**POPC-dHC** **POPC** **DPPC-dC**	POPC POPC-dH POPC-dHC	GC-MS, ^1^H-NMR, SANS	solvent-free and sucrose-free aLUVs prepared via mβCD-mediated exchange, hd-strategy	[62]
GUV	n.a.	**bSM** **24:1-SM** **16:0-SM** **18:0-SM**	POPC/SM POPC ± chol	TLC, FA (DPH, TMA-SPH)	mβCD-induced exchange (ha-strategy), influence of lipid composition on AChR distribution in symmetric and asymmetric liposomes	[55]
LUV	n.a.	**py-PG** **py-PG** **py-PI**	POPC	F (pyrene)	kinetic analysis of mβCD-mediated exchange via real-time monitoring of intervesicular lipid transfer	[64]
LUV	60% ^4^	**DPPC** **DPPC-d62** **POPC-d44** **POPC**	POPC POPC-d44 POPC-d31	GC-MS, UPLC-MS, ^1^H-NMR, SANS	SANS and SAXS analysis of aLUVs prepared via mβCD-induced lipid exchange, hd-strategy	[65]
LUV	59% ^4^	**DPPC-dC**	DPPC-dH	^1^H-NMR, GC	mβCD-mediated exchange (hd-strategy), lipid flip-flop in gel and fluid bilayers	[60]
LUV	0.48–0.67 ^7^	**POPC** **POPE**	POPE POPC	DSC, UPLC-MS, ^1^H-NMR	mβCD-induced exchange (hd-strategy), leaflet-specific lipid packing and melting	[66]
LUV, SUV	0.34–0.45 ^8^	**DMPC-d54** **eSM** **DPPC** **DPPC**	POPC-d13 POPC POPE ± chol	GC-MS, ^1^H-NMR	detailed protocol for the preparation of asymmetric vesicles via mβCD-mediated lipid exchange	[63]
LUV	≤85.9 ^4^	**mSM** **bSM** **eSM** **DMPC** **DPPC** **diC(15:0)PC** **DSPC**	DOPC ± chol	^1^H-NMR, HPTLC, FQ (Rho-DMPE, Rho-DOPE)	HPαCD-induced lipid exchange (ha-strategy), domain formation and interleaflet coupling using FRET	[81]
GUV, LUV	30–40% ^4^	**C24-SM** **C18-SM** **C16-SM** **C16-SM/C14-SM** **PC**	DPPC/DOPC/chol POPC/POPS/POPE POPC/POPS/DOPE	FM (NBD-DPPE, Rho-DPPE), MS FA (TMA-DPH)	mβCD-induced exchange (ha-strategy), influence of C24 sphingolipids on cholesterol and membrane microdomains	[67]
LUV	≤75% ^4^	**bSM**	DOPC sterol (chol, epichol, lanosterol, 7-dehydrochol, 4-cholesten-3-one)	FRET (Rho-DOPE, DPH), HPTLC	HPαCD-induced exchange (ha-strategy), incorporation of different sterol structures into aLUVs	[82]
LUV	0.32–0.45 ^9^	**POPC-d31** **DMPC-d54**	POPC POPC-d13	GC-MS, ^1^H-NMR	mβCD-mediated exchange (hd-strategy), influence of Gramicidin on lipid flip-flop and membrane- protein interactions	[68]
LUV	0.44 ^9^	**eSM**	POPE	^1^H-NMR, ^31^P-NMR	mβCD-induced lipid exchange (hd-strategy), studies of bending fluctuations via neutron spin-echo spectroscopy	[69]
LUV	7 mol% ^10^, 3 mol% ^5^	**POPC-d31**	POPS	GC-MS, F Annexin V assay (Annexin V-568)	mβCD-mediated exchange (hd-strategy), influence of PS asymmetry on the membrane interaction of pHLIP	[70]
LUV	≤100% ^8^	**POePC** **DOTAP** **POPS** **POPG** **POPA**	POPC POePC/POPC DOTAP/POPC POPS/POPC POPG/POPC POPA/POPC ± chol	HPTLC, F (DPH, TMA-DPH)	mαCD-induced lipid exchange (ha-strategy), entrapment properties of aLUVs containing one cationic and/or anionic leaflet	[83]
LUV	41–96% ^11^	**POPE/TOCL**	POPC	TNBS-labeling F (TTAPE-Me)	mβCD-mediated exchange with donor-SUVs instead of MLVs, effect of lipid asymmetry on MOM permeabilization by apoptotic proteins (tBid/Bax)	[89]
LUV	73% ^4^	**POPC**	POPE/POPS/chol	TLC, F (TMA-DPH)	mαCD-induced lipid exchange with CsCl entrapped in aLUVs	[84]
LUV	n.a.	**POPE**	POPG POPG-d31	UPLC-MS, GC	mβCD-mediated exchange (hd-strategy), interactions between aLUVs and frog peptides (L18W-PGLa, MG2a) or lactoferricin derivative LF11-215	[71]
LUV	55–70% ^4^	**MSPC** **SMPC** **PMPC** **MSM** **POPC** **SOPC**	DPPC	GC, SANS	mβCD-induced exchange (hd-strategy), transleaflet coupling of aLUVs in the fluid phase	[72]

^1^ Lipid conversion (in the outer leaflet); ^2^ a = asymmetry parameter, ranges from −1 (all-inside localization of PS) via 0 (symmetric distribution) to 1 (outside-only PS) [15]; ^3^ Total PC molecules located to the inner leaflet; ^4^ Outer leaflet lipids exchanged; ^5^ Amount of destined lipid located in the outer leaflet; ^6^ Component fraction in the outer versus inner leaflet; ^7^ ∑_as_ = degree of asymmetry, mole fractions of donor lipid in the outer and inner bilayer leaflets [66]; ^8^ Exchange efficiency: fraction of outer leaflet exchange [63]; ^9^ Mole fraction of incorporated lipid in the outer leaflet; ^10^ Amount of destined lipid located in the inner leaflet; ^11^ Lipid molar ratio between outer leaflet and total individual lipid content; ^12^ P^a%^ = percentage of lipid exchange in the asymmetric vesicle [45,46,47]; ^13^ Lipid transported; ^14^ a = asymmetric degree, based on PS monolayer concentration in the inner and outer leaflet [27]; Acronyms: C chromatography, F fluorescence, FA fluorescence anisotropy, FM fluorescence microscopy, FQ fluorescence quenching, FCS fluorescence correlation spectroscopy, FFE free-flow electrophoresis, FLIM fluorescence lifetime imaging, FRET Förster resonance energy transfer, GC gas chromatography, HPTLC high-performance thin-layer chromatography, M microscopy, MS mass spectrometry, NMR nuclear magnetic resonance spectroscopy, SANS small angle neutron scattering, TLC thin layer chromatography, UPLC ultra-performance liquid chromatography.

The method can be fine-tuned to fabricate bilayers of different intended compositions including a variety of acyl chain structures as well as lipid mixtures and cholesterol [61,73,74,76]. Note that asymmetric vesicles are hardly formed from lipids with two short or polyunsaturated acyl chains due to transverse diffusion [76]. Any change in lipid composition should first be examined with dynamic light scattering (DLS)to determine the mβCD concentration at which vesicles are dissolving [63]. Lin et al. [78] used HPαCD instead of mβCD to improve vesicle yields and control the amount of cholesterol introduced. However, difference in affinity of HPαCD for various lipids may lead to less efficient exchange [78]. Exchange efficiencies, defined as the fraction of outer leaflet exchange and calculated in consideration of the donor mole fraction, are about 0.35–0.45 [63]. If higher efficiency is desired, the donor/acceptor ratio can be increased, or multiple rounds of exchange can be performed. However, additional exchange steps may reduce vesicle yield [63]. In contrast, the presence of cholesterol significantly improves vesicle yield in many cases. Further, lipid charge affects exchange efficiency: using donor vesicles containing cationic phospholipids results in lower exchange efficiency, compared to using donor vesicles containing anionic phospholipids [83]. Donor lipid contamination can occur in the inner leaflet of acceptor vesicles in some cases [60,63]. Despite separation, residual aqueous cyclodextrin is still found in the system but has no measurable effects on lipid motion within the bilayer [60].

The extent of asymmetric insertion of a lipid from donor liposomes or bilayers (D1, D3, D4, D5) depends on time and the number, accessibility and composition of the donors. Let us, for example, assume one is aiming at exchanging 50 mol% of PC for PG in the outer leaflet of acceptor liposomes. One principal way would be to equilibrate the acceptor liposomes with the proper amount of donor liposomes. In our example, to reach 50% donor lipid, the accessible, outermost leaflet of the donors would need to include the same amount of lipid as that of the acceptor. Alternatively, one could equilibrate acceptors with a large excess of 50 mol%-PG donors. The rate of equilibration increases with cyclodextrin concentration (D5), donor curvature (D4) and temperature. If equilibration is too slow to be completed, more donor would be needed for the same result.

## 3. Testing Asymmetry

Establishing and, to a large degree, safely employing a protocol to establish lipid asymmetry requires a means to quantify asymmetry. The choice of a validation method is not connected to the preparation protocol and hence, not within the scope of this review. But to give an overview, we will briefly take a look at the different strategies.

An ideal method to quantify the transmembrane distribution of a single charged lipid is the precise measurement of the zeta potential. It utilizes the fact that the low dielectric permittivity of the membrane core renders the charges in the inner leaflet invisible without any additional quenching or labeling as needed for other methods. Zeta potential measurement provides a label-free and non-destructive assay for asymmetry verification. It should be noted that experience and non-standard procedures or accessories such as high-concentration or dip cells might be needed to ensure that the zeta measurement reaches the necessary precision.

Fluorescence is a very versatile technique and fluorescence can be quenched selectively in the outer leaflet. However, usually high amounts of quencher must be used, resulting in increased osmolarities outside the liposomes. Annexin-V assay is suitable for specific quantification of PS lipids in the outer leaflet. It is a very sensitive technique but the sample cannot be used for further analysis. Fluorescence anisotropy can be used for studies of lipid order. Other options include fluorescence microscopy, which is suitable solely for testing GUVs. 

It should be noted that the presence of fluorophores attached to lipids changes some of their properties crucially. This limits the applicability to asymmetric vesicles. For example, it essentially disqualifies fluorescent lipids from being proper models of unlabeled lipids for flip-flop studies.

NMR with shift reagents detects the fraction of a lipid species that is accessible from outside. Scattering techniques (SANS, SAXS) require or profit from deuterium labelling, which should be less intrusive to membrane properties than fluorophores. Mass spectrometry, optionally combined with other analytical methods such as gas chromatography, enables the analysis of molar masses, i.e. isotopically asymmetric labelled lipids. However, destruction of the sample has to be accepted.

## 4. Cholesterol

A membrane component outside the defined scope of lipids discussed here but of key interest for model membranes is cholesterol. Biological membranes, such as erythrocyte membranes, contain considerably larger amounts of cholesterol in their outer leaflet. Unlike the phospholipids discussed so far, cholesterol undergoes a fast flip-flop across the membrane so that its distribution is essentially equilibrated. Asymmetry is imposed by the facts that (I) the outer leaflet contains more unsaturated (sphingo)lipids with high cholesterol affinity than the inner and (II) the outer leaflet contains lesser phospholipid (intrinsic area) all together, giving way to the area requirement of additional cholesterol.

Together with mixing entropy opposing asymmetry, these properties give rise to cholesterol asymmetry [90,91]. It appears that creating sphingomyelin- and area asymmetry would be a means to establish a proper cholesterol asymmetry in a model membrane.

Two strategies should be possible to avoid cholesterol to interfere with cyclodextrin-based protocols for phospholipid asymmetry. First, α-cyclodextrins may be more challenging for lipid transfer but do not complex cholesterol. Hence, lipids can be handled selectively in the presence of cholesterol. Second, β-cyclodextrins bind cholesterol primarily with a stoichiometry of 2:1 but phospholipids with 4:1; given the binding constants, it needs relatively low cyclodextrin concentrations (of the order of 5–10 mM mβCD) to transport cholesterol [92] but higher cyclodextrin concentrations of about 30–50 mM to extract significant amounts of lipid [93]. Hence, one could in principle deal with the lipids first, at high cyclodextrin, and then add cholesterol using low cyclodextrin. The issue remains, though, that a relaxed lipid membrane with matching intrinsic areas of the outside and inside lipids will also incorporate cholesterol in a largely symmetrical manner to avoid asymmetry stress.

## 5. Outlook

Hoping the reader can forgive our roadmap story one more time, it needs to be reiterated that there is heavy construction going on in the country we reviewed. A new freeway or a new settlement in one area may suddenly redirect traffic and render Main Street a neglected place. In a foreseeable future, there shall be several alternative protocols available for every simple model of interest and time will have to tell which of them become standard. At some point, using symmetric vesicles for a model study may become as unpopular with reviewers as it happened with DMPC as generic membrane model some decades ago. More sophisticated types of asymmetry as exemplified for cholesterol will become accessible at some point.

From our perspective, it would be useful to agree on a uniform definition of lipid asymmetry in the future. The column regarding degree of asymmetry in Table 1. shows that 14 different definitions of asymmetry are present. To date, there have been a number of attempts to define lipid asymmetry more clearly. We previously introduced an asymmetry parameter *a* that comprises the amount of asymmetrically distributed lipid, i.e. PS, and ranges from −1 (all-inside localization of PS) via 0 (symmetric distribution) to 1 (outside-only PS) [15]. Guo et al. described the asymmetric degree *a* of PS molecules in the membrane based on its monolayer concentration in the inner and outer bilayer leaflet [27]. Eicher et al. defined asymmetry, ∑_as_, as the difference of donor lipid mole fraction in the outer and inner leaflet [66]. Another approach defines P^a%^ as percentage of lipid exchange in asymmetric vesicles with regard to symmetric and asymmetric vesicles [45,46,47].

The above mentioned, varying and in some cases lacking definitions of lipid asymmetry demonstrate the need for a uniform specification to improve comparability of the large number of methods for preparation of liposomes.

Another aspect of interest and potential for further investigation is the asymmetry stability issue. In individual cases, if specifically highlighted in the article concerned, we have already mentioned asymmetry stability in chapter two (see above). That topic receives varying levels of attention; while in some protocol asymmetry is confirmed for at least a few hours, others provide stability data over several days. Again, it is not trivial to compare data of different methods and systems. 

We tried our best to compile a selection of optimal protocols based on what the individual papers are providing. In the future, it will be important to have comparative reports of one laboratory having tested different protocols to serve a given purpose. This will give rise to improved, dedicated roadmaps for a vehicle of interest. 

We now slowly start getting rewarded for using asymmetric models by learning about the functions of lipid asymmetry in biology and its potential use for technical applications.

## Figures and Tables

**Figure 1 membranes-13-00267-f001:**
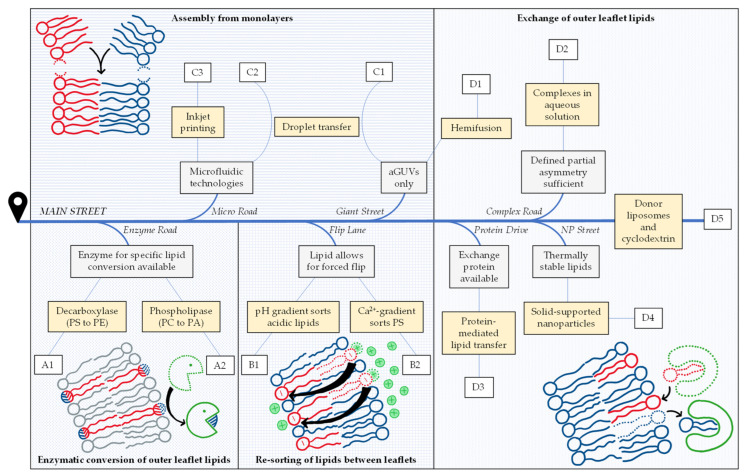
Road map to asymmetry, illustrating the criteria and considerations to select one or more suitable protocols described in the literature for preparing lipid-asymmetric vesicles that suit a given purpose. Eleven protocols are distinguished (yellow boxes) that can be grouped into four strategies (A–D). So far, exchange protocols represent the most abundant and most versatile strategy—illustrated here as the destination of Main Street. Gray boxes (see text for more detail) indicate criteria for exits to alternative protocols that may be favorable for a given purpose. Examples for each protocol (**A1**–**D5**) are listed in Table 1.

## Data Availability

Not applicable.

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
