# Peer review of "A Guide to Your Desired Lipid-Asymmetric Vesicles"

_membranes, 2023, doi:10.3390/membranes13030267_

Round 1

Reviewer 1 Report

With this manuscript, Mona Krompers and Heiko Heerklotz succeeded in compiling a large amount of information regarding the fabrication of asymmetric vesicles as well as giving an easy-to-follow comprehensive analysis of their fundaments, advantages, and drawbacks. I have no doubt that this revision work will be of broad interest to the readers of “Membranes”.

I have only one suggestion that may, in some way, improve the work. In my opinion, the asymmetry stability issue is one of the more important drawbacks to the wider use of asymmetric vesicles as regular membrane models. This issue remains somehow discussed elusively throughout the manuscript and I understand that it deserves a more highlighted place, maybe in the same way that is treated “testing asymmetry” or “cholesterol” content in sections 3 and 4. It would help the overall work if the authors add a short section on this issue.

Author Response

Dear reviewer,
we are impressed by the speed of the review procedures and grateful for your very positive view and constructive comments.

We agree that effort and reliability of a protocol and stability of the liposomes are important selection criteria and we have cited some information on these points from the original papers. Unfortunately, we could not use them for a systematic ranking of the protocols for a given purpose. To explain this problem, we have added the following paragraph to the „Navigating“ section:

"For your choice, you will need to rank your options to optimally suit your purpose with respect to the degree of asymmetry, reliability, stability, absence of disturbing components and the time, equipment, materials and expertise needed. Our review cannot solve this problem for you. We are citing some available information on these points given in the original papers but we are lacking a true comparison of different protocols done by the same lab. Furthermore, the best choice depends on the problem and the equipment and experience of a given laboratory. In other words, we try to provide a map but your best path will depend on whether you are driving a racecar, a 4x4 or a bicycle."

In the outlook, we highlighted the need for comparative reports:

"We tried our best to aid a selection of optimal protocols based on what the individual papers are providing. In the future, it will be important to have comparative reports of one laboratory having tested different protocols to serve a given purpose. This will give rise to improved, dedicated roadmaps for a vehicle of interest."

In line with these points, also the problem of whether the preparation protocol affects the stability of asymmetry is not solved but a topic for the outlook. We have added:

"Another aspect of interest and potential for further investigation is the asymmetry stability issue. In individual cases, if specifically highlighted in the article concerned, we have already mentioned asymmetry stability in chapter two (see above). That topic receives varying levels of attention: While in some protocol asymmetry is confirmed for at least a few hours, others provide stability data over several days. Again, it is not trivial to compare data of different methods and systems."

Reviewer 2 Report

The work describes the state of the art on asymmetric membrane vesicles manufacturing approach.

It is very well organised, well written and very funny to read based on the very nice way, the authors set up the manuscript.

I would only suggest to add some comments about the reproducibility of results of each method, which method is more reliable compared to the others. 

Moreover, which method can be used in a easier way and which one is more laborious. 

Author Response

Dear reviewer,
we are impressed by the speed of the review procedures and grateful for your very positive view and constructive comments.

We agree that effort and reliability of a protocol and stability of the liposomes are important selection criteria and we have cited some information on these points from the original papers. Unfortunately, we could not use them for a systematic ranking of the protocols for a given purpose. To explain this problem, we have added the following paragraph to the „Navigating“ section:

"For your choice, you will need to rank your options to optimally suit your purpose with respect to the degree of asymmetry, reliability, stability, absence of disturbing components and the time, equipment, materials and expertise needed. Our review cannot solve this problem for you. We are citing some available information on these points given in the original papers but we are lacking a true comparison of different protocols done by the same lab. Furthermore, the best choice depends on the problem and the equipment and experience of a given laboratory. In other words, we try to provide a map but your best path will depend on whether you are driving a racecar, a 4x4 or a bicycle."

In the outlook, we highlighted the need for comparative reports:

"We tried our best to aid a selection of optimal protocols based on what the individual papers are providing. In the future, it will be important to have comparative reports of one laboratory having tested different protocols to serve a given purpose. This will give rise to improved, dedicated roadmaps for a vehicle of interest."

Reviewer 3 Report

The review is well written and conceived. The authors reported in-vitro methods that have been established to prepare asymmetric liposomes (lipid model membranes). It resulted quite interesting and of importance for a great audience working with lipid membranes. The paper is publishable as it is. 

Author Response

Dear reviewer,

we are impressed by the speed of the review procedures. Thank you very much for your very positive assessment.